REGISTERED REPORT PROTOCOL

# Evaluating the ability of an artificial-intelligence cloud-based platform designed to provide information prior to locoregional therapy for breast cancer in improving patient's satisfaction with therapy: The CINDERELLA trial

Orit Kaidar-Person[1,2]*, Marilia Antunes[3,4], Jaime S. Cardoso[5,6], Oriana Ciani[7], Helena Cruz[8], Rosa Di Micco[9], Oreste D. Gentilini[9], Tiago Gonçalves[5,6], Pedro Gouveia[8,10], Jörg Heil[11], Pawel Kabata[12], Daniela Lopes[8], Marta Martinho[8], Henrique Martins[8,13], Carlos Mavioso[8], Martin Mika[14], Helena Montenegro[5,6], Helder P. Oliveira[5,15], André Pfob[11], Nicole Rotmensz[9], Timo Schinköthe[14], Giovani Silva[3,4], Rosana Tarricone[7], Maria-Joao Cardoso[8,10], on behalf of the CINDERELLA Consortium[¶]

1 Breast Cancer Radiation Therapy Unit, Sheba Medical Center, Ramat Gan, Israel, 2 School of Medicine, Tel-Aviv University, Tel-Aviv, Israel, 3 Faculdade de Ciências, Universidade de Lisboa, Lisboa, Portugal, 4 Instituto Superior Técnico, Universidade de Lisboa, Lisboa, Portugal, 5 Center for Research on Health and Social Care Management (CERGAS), SDA Bocconi University, Milan, Italy, 6 Champalimaud Foundation, Breast Unit, Lisbon, Portugal, 7 Institute for Systems and Computer Engineering, Technology and Science (INESC TEC), Porto, Portugal, 8 Faculty of Engineering, University of Porto (FEUP), Porto, Portugal, 9 Breast Surgery Unit, San Raffaele University and Research Hospital, Milano, Italy, 10 Faculdade de Medicina da Universidade de Lisboa, Lisboa, Portugal, 11 Instituto Universitario de Lisboa (ISCTE), Lisboa, Portugal, 12 CANKADO GmbH, Ottobrunn, Germany, 13 Centro de Estatística e Aplicações, Faculdade de Ciências, Universidade de Lisboa (CEAUL), Lisboa, Portugal, 14 Department of Surgical Oncology, Faculty of Medicine, Medical University of Gdańsk, Gdańsk, Poland, 15 Department of Obstetrics & Gynecology, Heidelberg University Hospital, Heidelberg, Germany

¶ Membership in the CINDERELLA consortium is provided in the Acknowledgments.
* orit.kaiderperson@sheba.health.gov.il

## Abstract

### Background

Breast cancer therapy improved significantly, allowing for different surgical approaches for the same disease stage, therefore offering patients different aesthetic outcomes with similar locoregional control. The purpose of the CINDERELLA trial is to evaluate an artificial-intelligence (AI) cloud-based platform (CINDERELLA platform) vs the standard approach for patient education prior to therapy.

### Methods

A prospective randomized international multicentre trial comparing two methods for patient education prior to therapy. After institutional ethics approval and a written informed consent, patients planned for locoregional treatment will be randomized to the intervention (CINDERELLA platform) or controls. The patients in the intervention arm will use the newly designed web-application (CINDERELLA platform, CINDERELLA APProach) to access the

**Data Availability Statement:** All data (anonymized patient data; image data) not subject to any IPR, GDPR or security rules restrictions will be uploaded in a publicly available format such as the open-access repository Zenodo (https://zenodo.org) – an Open Data Commons licensing will be adopted using proper unique DOI indexing.

**Funding:** Funding: EU grant HORIZON-HLTH-2021-DISEASE-04-04. HORIZON-RIA. Proposal number 101057389 The funders did not and will not have a role in study design, data collection and analysis, decision to publish, or preparation of the manuscript.

**Competing interests:** The authors have declared that no competing interests exist.

information related to surgery and/or radiotherapy. Using an AI system, the platform will provide the patient with a picture of her own aesthetic outcome resulting from the surgical procedure she chooses, and an objective evaluation of this aesthetic outcome (e.g., good/fair). The control group will have access to the standard approach. The primary objectives of the trial will be i) to examine the differences between the treatment arms with regards to patients' pre-treatment expectations and the final aesthetic outcomes and ii) in the experimental arm only, the agreement of the pre-treatment AI-evaluation (output) and patient's post-therapy self-evaluation.

## Discussion

The project aims to develop an easy-to-use cost-effective AI-powered tool that improves shared decision-making processes. We assume that the CINDERELLA APProach will lead to higher satisfaction, better psychosocial status, and wellbeing of breast cancer patients, and reduce the need for additional surgeries to improve aesthetic outcome.

## Introduction

Shared decision-making is a key component of patient-centred health care. This is a process in which the care team, particularly the treating physician and the patient, work together to make decisions on the treatment plans based on clinical evidence and the individual clinical case, while balancing risks and expected outcomes with patient preferences, expectations, moral beliefs, and values [1]. The physician's role is to clearly communicate the pros and cons of each approach, the risks, benefits and consequences of each approach, in a language and manner that the patient can easily comprehend. This approach empowers patients to make informed decisions about the proposed options based on a broad understanding of each treatment offered. However, providing information is often not sufficient to help the patient achieve full appreciation of what is at stake and give his/her informed consent [1].

Newly diagnosed breast cancer patients are put in the position of having to make difficult decisions several times during their treatment journey [2]. Current treatment approaches are multimodal, including surgery, systemic therapy, and radiation [3]. Patients might be given a choice for sequencing therapy (e.g., systemic therapy before or after surgery) that may influence the type of locoregional therapy (e.g., from mastectomy to breast conservation). In each modality, the patients might be offered different choices of therapy (e.g., different systemic agents, mastectomy or oncoplastic surgery), based on their personal risk factors (e.g., age, comorbidities) and disease-related factors (e.g., stage, molecular subtype) and physician preference. Surgical management is no longer the one-size-fits-all mutilating procedure of radical (even if modified) mastectomy that governed breast surgery for almost 100 years, but evolved to offer better aesthetic outcomes, improved quality of life (QoL) and psychological well-being of patients [4, 5]. For a given breast cancer stage, the patient might be offered different types of breast and lymphatic surgeries and various radiation protocols, with similar oncological outcomes, which might be challenging to explain to the patient [2, 5]. Based on the patient's individual risks/factors (e.g., smoking, body mass index), location of the primary tumour, surgical procedure and surgeon's expertise, and whether radiation was applied, the aesthetic outcomes might vary significantly between patients undergoing the same treatment [6].

Current practice for deciding on locoregional therapy and shared decisions varies between centres and may include pictures, drawings, brochures, referral to the centre's website, and a

meeting with a variety of staff members, but it is not uniform and does not consider all the patient's individual factors. In addition, breast cancer patients who refer to social media or unsupervised websites for information may be subjected to misinformation [7]. Social media can be particularly persuasive as it is used daily to access a wide range of information including medical information. Moreover, the information has an emphasis on visual content and personal experiences [7]. Thus, this misinformation might challenge the shared decision process of the patient and her caregiver.

Even though locoregional therapy has significantly improved, one third of the patients might have poor aesthetic outcomes, which in turn may also affect patient's well-being [8]. Furthermore, the different treatment options are based on knowledge of oncological outcomes and there are no tools currently available for everyday clinical practice use to help predict or evaluate patient-centred outcomes, in particular aesthetic outcomes. Quality indicators have not been developed neither for the aesthetic evaluation of locoregional breast cancer treatments, nor for measuring patient's satisfaction and matching of expectations [9]. Patients might value outcomes differently when comparing them with what they expected [10]. This helps explain why this important aspect of breast cancer care is poorly standardized and is often out of scope when auditing procedures. This means that in many cases of "shared decision process" for locoregional therapies in breast cancer, both the caregiver and the patient are left unguided and cannot fully anticipate the impact of each therapy on the aesthetic outcome, patient's body image, well-being, and QoL. Moreover, the process might be challenged by prejudice due to misinformation from unsupervised sources of information [1, 7].

The CINDERELLA project is based on collaboration between leading breast cancer clinical centres and health industry partners INESC TEC & CANKADO to create an easy-to-use, cost-effective web-based tool for patients and their families to improve the communication with health providers and assist the informed consent process prior to locoregional therapy for breast cancer. As part of the project, the CINDERELLA trial is planned as a prospective multi-centre randomized controlled trial [Clinical Trials.gov Identifier 05196269] to evaluate conventional methods for patient education prior to therapy versus an artificial-intelligence cloud-based healthcare platform (CINDERELLA APProach) in breast cancer patients planned for locoregional treatment. The project is supported by the European Union [EU grant HORI-ZON-HLTH-2021-DISEASE-04-04. HORIZON-RIA. Proposal number 101057389]. The interventional arm will have access to a newly design web-platform (CINDERELLA platform, CINDERELLA APProach), that provides information about the locoregional therapy and the possible aesthetic outcomes (visual with a scale) according to the different type of surgeries the patient was offered. The types of surgeries offered, and patient's choice will be recorded for both groups. Patients' expectation and satisfaction with the outcomes will be recorded. We anticipate that the CINDERELLA APProach will improve patient's expectations and result in greater satisfaction with locoregional therapy outcomes compared to methods currently used in clinical practice.

## Methods

Prospective randomized international multicentre parallel group trial conducted in 5 clinical centres, each attaining an institutional ethics approval. All patients will need to sign a written informed consent form. Randomization will be performed centrally, 1:1 to control arm (conventional current clinical practice for patient education prior to therapy) or intervention arm. Randomization will follow the minimization method proposed by Pocock and Simon [11, 12], a covariate-adaptive randomization procedure that achieves balance within covariate margins. The randomization procedure requires randomization lists of two types: (1) 50:50 lists, with

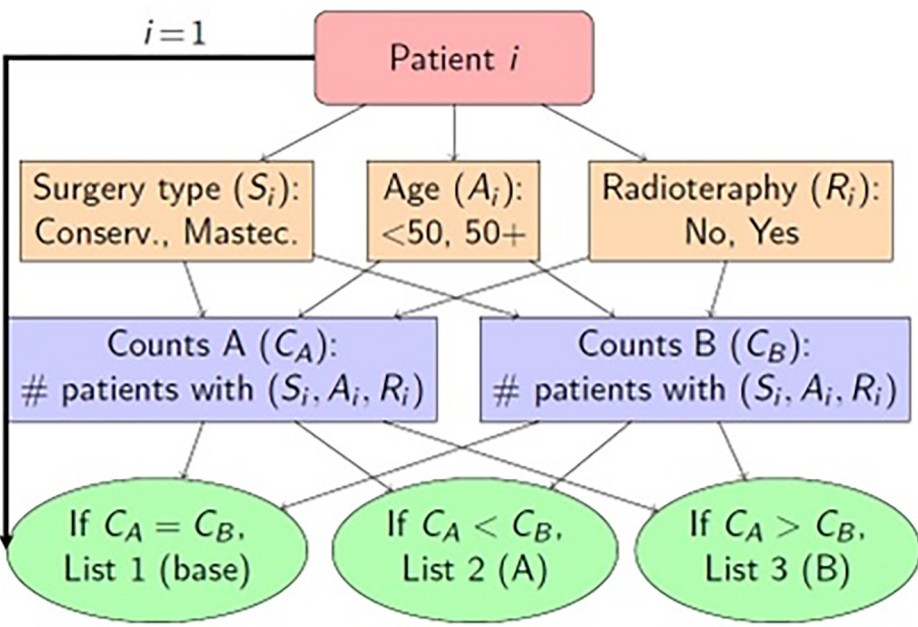

**Fig 1. Rationale of the randomization method.**

blocks of size two will be used to assign the patients when balance between groups is attained during the randomization process. (2) Unbalanced 20:80 and 80:20 randomization lists in blocks of size five will be used to assign the patients otherwise (Fig 1).

The intervention arm will have access to the CINDERELLA platform which entails the information related to surgery and/or radiotherapy. Upon patient's request, the platform will provide the patient with a matched picture as an example of her own aesthetic outcome according to the surgical procedure she chooses, and an objective evaluation of this aesthetic outcome (e.g., good/fair). The trial will not interfere with the type of locoregional therapy offered to the patient, or any treatment, but will evaluate the CINDERELLA approach and standard approach in terms of achieving an optimized shared decision making.

In total, considering both primary objectives (i) to evaluate the differences between the arms in patient's agreement of expectations and aesthetic outcome and (ii) in the experimental arm only, the agreement of the AI-evaluation and patient's post-therapy self-evaluation, it is estimated that a minimum of 515 patients will be enrolled in each arm of the study, based on recent information (frequency and dropout) from the clinical centres. The trial objectives are summarized in Table 1. Sample size was calculated in a two-step simulation-based approach using information from a previous study and expert clinical opinion for the 1st and the 2nd primary objectives, respectively. The simulation studies considered the use of weighted Cohen's k to calculate the level agreement with a precision of 0.1 and a confidence level of 95% [13, 14]. The proposed sample size was obtained after a 40% correction to account for potential dropout.

Stratification will be considered to balance potential confounding factors that might influence aesthetic outcomes, i.e., type of surgery (conservative, mastectomy), patient age (<50, 50 +) and radiotherapy (yes, no) (see flowchart in Fig 1). This randomization procedure was implemented and simulated using R, a free software environment for statistical computing and graphics (R Core Team, 2022 ⓇⓇ), following the minimization method proposed by Pocock and Simon [11, 12], a covariate-adaptive randomization procedure that achieves

**Table 1. CINDERELLA trial–objectives.**

| TRIAL OBJECTIVES |
| --- |
| **Primary objectives:** |
| 1. To evaluate the difference between both arms of the trial, regarding the match of expectations about the aesthetic outcome before and after treatment. |
| 2. Experimental arm only, the agreement of the pre-treatment AI-evaluation (output) and patient's post-therapy self-evaluation. |
| **Secondary objectives:** |
| 1. To evaluate the difference, between both arms of the trial, in patient's body image satisfaction after surgery. |
| 2. To evaluate the difference between both arms of the trial, in health resource consumption (time spent in hospital, number of appointments, duration until treatment additional care sought by patients). |
| 3. To evaluate the difference, between both arms of the trial, in patient's satisfaction with their professional life. |
| 4. To evaluate the difference between both arms of the trial, in patient's satisfaction with their sexual life. |
| 5. To evaluate the difference, between both arms of the trial, in patient's satisfaction with their general health-related quality of life. |

balance within covariate margins. The randomization was designed to be implemented in each of the centres and then, integrate the trial as a whole.

Eligibility criteria include: Female aged more than 18 years old; written informed consent; primary breast cancer in situ or invasive without evidence of systemic disease (non-Stage IV or locally advanced non-operable breast cancer); ECOG performance status 0 or 1; Uni or Bilateral breast surgery even if prophylactic in one side; Ability to use a web-based app autonomously or with home-based support. Exclusion criteria include: Mastectomy without reconstruction; pregnancy or lactation; Previous radiation to breast/chest (e.g., lymphoma); previous ipsilateral breast surgery due to malignant disease; other neoplasm in the last 5 years (excluding basal cell carcinoma of the skin and adequately treated carcinoma in situ of the cervix); severe skin disease that will contra-indicate the use of radiotherapy; previous breast prophylactic surgery; uncontrolled psychiatric disorder.

After randomization the patient will either follow the investigational arm with the CINDERELLA APProach vs. a conventional approach (control arm). For both study arms, patient's breasts will be photographed (without identifiers) prior to any intervention. The patients will fill in an expectations questionnaire, resource consumption survey and patient-reported outcome measures (PROMs): EQ-5D-5L, EORTC QLQ C-30 and EORTC BR-23/ EORTC QLQ BR-45, Harris Scale, and BREAST Q ICHOM. Questionnaires and photographs will be repeated after healing is completed, six months and one year after the end of treatment (surgery or radiotherapy if radiotherapy was done). Data manager will evaluate that all collected data and questionnaires are completed properly at the time of submission by the centres, to reduce the possibility of missing information and non-compliance with the protocol.

Investigational arm-only: prior to deciding on locoregional therapy the patient will be given access to the CINDERELLA platform. The platform will contain clinical information as mentioned above. The patient breasts will be photographed prior to therapy by the research team. The BreLO AI (version 1.0) will match the patient biometrics and images with a similar case existent in the BreLO repository and already classified by the BCCT.core (version 4.0) into excellent, good, fair and poor. The patient can then visualize the results presented as synthetized images of her own body. BreLO AI will include an automatic image quality assessment module; if the images of the patient are below a quality standard, the matching will fail with an error message for the user.

All trial information will be fed into the platform to improve the AI-system for performing biomatching. To promote participant retention and complete follow-up, an intense training of nurses in each site as well as continuous proactive follow-up will be ensured. If participants discontinue or deviate from intervention, we will document that event but not obtain any special outcome data.

## Image acquisition protocol & innovative photo robot

The trial protocol includes image acquisition protocol to minimize variability of patient's imaging. The clinical partners will have an option to use a "Photo Robot" which is a special photo booth that includes a standardized setting for image acquisition and uploads an anonymized image to the CINDERELLA-platform. The Photo Robot is a prototype manufactured especially for the trial by a team with extensive experience in multiple photographs of complex and small sized objects and the reconstruction of 3D imagery.

## Statistical analysis

Once the data set is collected, an initial treatment ("cleaning") will be performed to eliminate potential inconsistencies, e.g. poorly completed questionnaire items, a broad characterization taking into account the dropout of patients or censored observations, and a comprehensive descriptive analysis of the data, namely looking for patterns in the data and corresponding hypotheses of interest for the inferential analysis.

For the primary objectives of the trial, models of the class of generalized linear models (in particular, multinomial regression models for ordinal data) will be fitted in order to evaluate the effect of the training and the women's characteristics on their evaluation of the aesthetic outcome of the surgery. In order to assess the improvement in the ability to classify the aesthetic result of their surgery provided by training, Weighted Cohen's k will be calculated for both groups (train and control) and compared using a statistical test and/or bootstrap techniques. A measure of similarity between self-evaluation and the BCCT.core will be computed for each participant and a beta regression model will be fitted to assess the effect of training, controlled by variables that can play as confounders such as women's and disease's characteristics at each time point and in a longitudinal perspective.

Analysis of the secondary objectives: (i) Patient's body image satisfaction after surgery; (ii) Resource consumption (time spent in hospital, number of appointments, duration until treatment, additional care sought by patients) and $CO_2$ emissions in both the intervention and the control arm; (iii) Patient's satisfaction with their professional life; (iv) Patient's sexual life satisfaction; (v) Patient's general health-related quality of life, the patient-reported outcome measures administered will be scored according to the official guidelines provided by the developers of the instruments. Utility values will be derived from the preference-based EQ-5D-5L questionnaire, based on the value set recently derived from a representative sample of the Italian population [15]. In this context, the collected data will also be analysed in an exploratory way after a potential cleaning and an elimination of inconsistencies. Unitary costs will be expressed as EUR 2023. Unitary costs will be derived from national price listings or in patient official tariffs, laboratory and instrumental tests will be valued according to the outpatient procedures formulary in accordance with the ongoing policies at the five centers. Taking account of the type of outcomes under study, adequate statistical analyses will be carried out again using generalized linear mixed models that are implemented in several R-specific packages ® [R Core Team, 2022]. All the statistical analysis will be conducted using R statistical software. The longitudinal character of some outcomes will allow estimating their profiles by detecting trends. All processes and analyses will be clearly documented for reproducibility purposes.

## Data collection, management and analysis

The clinical report forms (CRFs) will include patient's and treatment related information that will be used to improve the biomatching process. All data will be collected through the clinical registries and then uploaded at the CINDERELLA platform, Table 2 summarized timeline of data collection.

**Table 2. Timeline of data collection.**

| TIME LINE DATA COLLECTION | | | | |
|---|---|---|---|---|
| | Inclusion | Healing* | 6 months after end of treatment** | 12 months after end of treatment** |
| Signed informed consent/ Randomization | X | | | |
| EQ 5D; EORTC QLQ C30 BR-23 | X | X | X | X |
| BREAST Q ICHOM Pre and Post | | | | |
| Patient Data | X | X | X | X |
| Expectations | X | X | X | X |
| Picture Capture | X | X | X | X |
| BCCT.core | | | X | X |
| Resource consumption survey | X | | X | X |

*This point will be checked by the healthcare team as all wounds closed and scarring is complete or before the start of radiotherapy in patients proposed for radiotherapy

** Consider 6 and 12 months after healing when no radiotherapy was performed or 6 and 12 months after radiotherapy

Plans for data entry, coding, security, and storage, are very strict; they include the use of trusted cloud sources and experienced partner. Processes to promote data quality were discussed and a glossary as well as a wizard will be used to help guide manual data entry where applicable. Details of data management procedures can be found in the project Data Management Plan (Deliverable D.6.8/D6.9 –sensitive/confidential level). Data will be stored in the ISO27001 certified CANKADO Software. All clinical centres received institutional security approval for data safety of the whole trial process.

Data management strategy of CINDERELLA trial is planned according to FAIR (Findable, Accessible, Interoperable, and Reusable) principles. At regular intervals planned meetings with data management, information technology and data protection, and ethics board experts' representatives from relevant stakeholders will be held. This will serve to assure that data collected and shared by the consortium members and all analysis of personal data is performed according to the appropriate legal basis (General Data Protection Regulation for the EU GDPR, and corresponding legal basis in Israel). All information, including patient's photographs, will undergo de-identification processes and security measures as required by mentioned data protection regimes, and guidelines, as well as according to local policies and ethical sensitivities. The CINDERELLA APProach will be tested at different times and levels to check for its alignment with data protection regulation, ethical principles in usage of AI and bioethical standards/policies within the participating entities of different countries.

The EU team will monitor the inclusion rate and data completion along the clinical trial timeline. Reports will be available periodically to all clinical facilities to allow to correct deviations in a timely manner.

There is no direct human–AI interaction in the handling of the input data, in the sense that images to be collected as well as clinically relevant information are obtained from patients and from health professionals via the photo robot or by manual input into interfaces that do not run AI in the interaction process. Average digital skills are expected from users, but if the patient cannot document some of the data about herself, she can ask for support from family/caregiver. Regarding image capture—this will be facilitated by the automation involved, but the handling of the robot does not require advanced digital skills, nor does it imply direct human-AI interaction.

## Ethics, dissemination and patient association engagement

Ethics committee approval will be obtained prior to recruitment in each of the 5 clinical centres.

The consortium has a clear dissemination plan to promote the trial. The Cinderella website (www.cinderellaproject.eu) and social media accounts (Twitter, Facebook and Linkedin) will be used to disseminate the start inclusion point in all participating clinical centres through a vast net of contacts. Breast Cancer Advocacy Organizations (BCAO) will be tackled across Europe to inform patients about the trial sites. A promotion slide will be added to consortium member presentations in scientific meetings. In the first trimester of 2023 a meeting with BCAO from around Europe will be held in Lisbon to not just disseminate but evaluate dissemination messages and materials and work to co-create them and adjust some of the key messages and materials. All data (anonymized patient data; image data) not subject to any intellectual property rights, GDPR or security rules restrictions will be uploaded in a publicly available format such as the open-access repository Zenodo (https://zenodo.org)–an Open Data Commons licensing will be adopted using proper unique DOI indexing.

## Discussion

### 1. Previous unmet clinical needs

Female breast cancer has surpassed lung cancer as the most commonly diagnosed cancer, with an estimated 2.3 million new cases yearly (11.7%) [16]. Approximately 90% of all newly diagnosed breast cancer patients will undergo locoregional treatment (surgery and radiotherapy) and many of them will attain long-term survival. The continuous improvements in breast cancer treatments have led to a progressive reduction in breast cancer mortality accounting for 2–4% per year. As a result, approximately 8 million women are now living with a diagnosis of breast cancer in the past five years (https://www.who.int/news-room/fact-sheets/detail/breast-cancer). Once the oncological pathway is concluded, the only physical sign of the previous treatment is the aesthetic result of surgery and radiotherapy. Whilst breast cancer treatment period accounts for a relatively limited period of time in the total life of the patient, its consequences stay forever and she will be faced with the aesthetic outcome of surgery every day in the mirror. This has been shown to impact on patient's wellbeing and self-esteem thus affecting her quality of life after recovery. Women with early-stage disease are often given different treatment options that have similar disease outcome but might significantly affect their quality of life. There are more disability-adjusted life years (DALYs) lost by women to breast cancer globally than any other type of cancer [17]. Although improving survival is crucial, health related quality of life (HRQoL) should parallel this endpoint. HRQoL is very much dependent on the side effects of treatments including systemic treatments (chemotherapy, targeted therapies) and locoregional treatments (surgery and radiotherapy). In the last decade, medicine is evolving towards a more and more patient-centred approach where patient satisfaction and HRQoL are the driving forces along the care pathway. However, when poor aesthetic results occur after locoregional treatment, they disappoint patient's preoperative expectations leading her to psychosocial distress [18]. Aesthetic outcomes are difficult to predict and to communicate to the patient based on current tools (e.g., pictures), thus limiting the ability to have a "shared decision" or even an appropriate decision for an individual patient [9]. The introduction of AI in medical technologies has provided new solutions to improve patient management. AI includes machine learning that uses algorithms and data to mimic the way humans learn [19, 20]. AI is now an integral part of our daily living. Technology is implemented in many aspects of our lives and in some, helps to overcome the boundaries of socioeconomic differences and communication skills. The potential applications of AI in breast surgery vary from screening and diagnosis to prediction models and decision support systems. Current literature provides sufficient groundwork to build up randomized prospective trials studying the impact of AI in clinical practice [21]. Table 3 summarizes the ultimate clinical need for the

**Table 3. The ultimate clinical need for the CINDERELLA APProach.**

| UNMET CLINICAL NEEDS |
| --- |
| • The absence of any quality control on the aesthetic outcomes of breast cancer locoregional treatment;<br>• The absence of any standardized routine form of evaluation of the aesthetic outcomes of breast cancer locoregional treatment;<br>• The absence of any standardized evaluation of patient's expectations about aesthetic outcomes, satisfaction with treatments, impact on professional and sexual life and on QoL;<br>• The absence of easy to access consistent information (digital system) where patients can consult the options of locoregional treatments proposed to her and their outcomes matched to their specific image and body metrics;<br>• The absence of quality indicators in the auditing processes of breast units related to aesthetic outcome and satisfaction of breast cancer locoregional treatment. |

CINDERELLA APProach. The CINDERELLA platform is planned to have different levels of information to match different level of education and needs of the patients. The AI system might be limited to races/ethnicities of patients from the clinical sites that are involved in the project. The project is designed to have input from different stakeholders to upgrade/improve/ adjust the platform to patient's needs. The partners aim to involve non funded partners to join the project to increase the diversity of the dataset and to have representation of all races/eth-nicities to assure that the platform serves all.

## 2. Innovation in CINDERELLA APProach

CINDERELLA APProach aims to use AI to simplify the options for locoregional therapies to the patient, make the information easy to understand and interactive at different levels of engagements and search of information. It will assure that all patients will have access to the uniform high-quality information, including data visualization using graphical representation of various treatment options and aesthetic outcomes and patient's individual image based on biometric profile. By using these data visualization tools, the CINDERELLA APProach will provide an accessible way for the patient to fully understand treatment outcome. We aim that the CINDERELLA APProach will improve the shared decision process, and therefore, improve the lives of millions of breast cancer patients around the world. Furthermore, innovation key points of the CINDERELLA trial are the electronic collection of PROMs via CANKADO and the use of a large database of previously treated patients with biometric data, pre- and post-treatment images and aesthetic classification of results. For the first time, the CINDERELLA APProach will merge two different perspectives of QoL, integrating objective AI-powered methods and significant factors derived from patient's input. This project will design and implement a multidimensional health technology assessment, investigating the organizational performance, impact and sustainability of a new AI-based healthcare tool which depicts an innovative care pathway for patients. Additionally, CINDERELLA is an example of interdisci-plinarity and trans-disciplinarity due to the combination of resources from clinical and techni-cal areas like informatics, engineering and bioengineering, sharing the same objectives.

## 3. Expected results and impact

On the patients' side, the CINDERELLA APProach will enable them to futurize their results contributing to an increased awareness of the therapeutic pathway, a better-informed choice of the surgical procedure and, potentially, an improved QoL following a greater matching of expectations and results.

On the health care professionals' side, the CINDERELLA APProach will improve the com-munication with the patients as well as providing a digital platform with useful information for auditing purposes and quality indicators for breast units.

**Table 4. Key expected results of the CINDERELLA trial.**

| EXPECTED RESULTS |
|---|
| • Patients' expectations about locoregional treatment will be more frequently met. |
| • Less need for additional surgeries to improve aesthetic outcome. |
| • Higher satisfaction and better psychosocial wellbeing. |
| • Improvement in QoL. |
| • High quality information with reduced number of visits to the hospital lessening $CO_2$ emissions and costs without diminishing the quality of care. |
| • Multidisciplinary decisions can be supported by the patient's option and visualization of expected outcomes |
| • A defined standard for the evaluation of locoregional treatment can be used as a quality indicator for auditing purposes. |
| • Third party payers like insurance companies can be involved in decisions and support of payments that weren't until now contemplated due to the lack of outcomes evaluation. |

Furthermore, it is expected that the CINDERELLA TRIAL will impact on:

- scientific community where a new gold-standard evaluation of the breast appearance will be introduced;

- economy, as better matching of patient's expectations, higher satisfaction and improved aesthetic outcome will translate in a decreased number of secondary surgeries; and a digital-based approach will reduce the need of visits;

- environment, reducing $CO_2$ emissions with less visits to the hospital and the need for printed information;

- less developed countries where digital information could provide an easier and equable access to information.

We expect that after the end of this project the CINDERELLA APProach could be ideally adopted in all certified breast units so that thousands of women may benefit from a reduced rate of poor aesthetic results. Moreover, CINDERELLA results will highlight how digital technologies can improve lives providing further evidence on the trustworthy and ethical use of AI in health care.

Key expected results are summarized in Table 4.

## Supporting information

**S1 Checklist.**
(DOCX)

**S1 File.**
(DOCX)

## Acknowledgments

To our patients who inspire us to do better. All the members of the CINDERELLA consortium.

## Declaration

All experimental protocols were approved by the ethics committees at Sheba Tel Hashomer hospital, Champalimaud Foundation hospital, Gdańsk university hospital, San Raffaele hospital Ethics and Heidelberg university hospital. The project itself was approved by European's Commission Ethics Committee. Recruitment is planned to start at July 2023, in clinical sites

that achieved ethics approval. Written informed consent is mandatory for participation in the trial.

## Author Contributions

**Conceptualization:** Henrique Martins, Maria-Joao Cardoso.

**Data curation:** Marilia Antunes, Jaime S. Cardoso, Oriana Ciani, Helena Cruz, Rosa Di Micco, Oreste D. Gentilini, Tiago Gonçalves, Pedro Gouveia, Daniela Lopes, Marta Martinho, Henrique Martins, Helena Montenegro, André Pfob, Timo Schinköthe, Rosana Tarricone, Maria-Joao Cardoso.

**Formal analysis:** Oriana Ciani, Tiago Gonçalves, Henrique Martins, Helder P. Oliveira, Nicole Rotmensz, Timo Schinköthe, Giovani Silva.

**Investigation:** Orit Kaidar-Person, Jaime S. Cardoso, Rosa Di Micco, Jörg Heil, Pawel Kabata, Martin Mika, Helena Montenegro, André Pfob, Timo Schinköthe, Giovani Silva, Rosana Tarricone, Maria-Joao Cardoso.

**Methodology:** Marilia Antunes, Oriana Ciani, Pawel Kabata, Marta Martinho, Henrique Martins, Carlos Mavioso, Martin Mika, Nicole Rotmensz, Timo Schinköthe, Giovani Silva, Maria-Joao Cardoso.

**Project administration:** Helena Cruz, Pedro Gouveia, Daniela Lopes, Marta Martinho, Henrique Martins, Carlos Mavioso, Maria-Joao Cardoso.

**Software:** Jaime S. Cardoso, Tiago Gonçalves, Martin Mika, Helena Montenegro, Helder P. Oliveira, Nicole Rotmensz, Timo Schinköthe.

**Supervision:** Oriana Ciani, Helena Cruz, Oreste D. Gentilini, Pedro Gouveia, Jörg Heil, Pawel Kabata, Daniela Lopes, Henrique Martins, André Pfob, Giovani Silva, Rosana Tarricone, Maria-Joao Cardoso.

**Validation:** Henrique Martins, Martin Mika, Helder P. Oliveira, Maria-Joao Cardoso.

**Writing – original draft:** Orit Kaidar-Person.

**Writing – review & editing:** Orit Kaidar-Person, Marilia Antunes, Jaime S. Cardoso, Oriana Ciani, Helena Cruz, Rosa Di Micco, Oreste D. Gentilini, Tiago Gonçalves, Pedro Gouveia, Jörg Heil, Pawel Kabata, Daniela Lopes, Marta Martinho, Henrique Martins, Carlos Mavioso, Martin Mika, Helena Montenegro, Helder P. Oliveira, André Pfob, Nicole Rotmensz, Timo Schinköthe, Giovani Silva, Rosana Tarricone, Maria-Joao Cardoso.

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
