## [Decision Letter · Decision Letter 0]

2 May 2023

PONE-D-23-07134Evaluating the ability of an artificial-intelligence cloud-based platform designed to provide information prior to locoregional therapy for breast cancer in improving patient’s satisfaction with therapy: The CINDERELLA trial

PLOS ONE

Dear Dr. Kaidar-Person,

Thank you for submitting your manuscript to PLOS ONE. After careful consideration, we feel that it has merit but does not fully meet PLOS ONE’s publication criteria as it currently stands. Therefore, we invite you to submit a revised version of the manuscript that addresses the points raised during the review process.

ACADEMIC EDITOR:  I recommend the authors to have their manuscript checked by a native English-speaking person for various grammar, spelling and punctuation errors. Please also pay attention to enumerate the lines and pages of the manuscript.

We look forward to receiving your revised manuscript.

Kind regards,

Murat Akand, MD, FEBU

Academic Editor

PLOS ONE

Journal Requirements:

Reviewers' comments:

Reviewer's Responses to Questions

**Comments to the Author**

1. Does the manuscript provide a valid rationale for the proposed study, with clearly identified and justified research questions?

Reviewer #1: Yes

Reviewer #2: Yes

2. Is the protocol technically sound and planned in a manner that will lead to a meaningful outcome and allow testing the stated hypotheses?

Reviewer #1: Yes

Reviewer #2: Yes

3. Is the methodology feasible and described in sufficient detail to allow the work to be replicable?

Reviewer #1: Yes

Reviewer #2: Yes

4. Have the authors described where all data underlying the findings will be made available when the study is complete?

Reviewer #1: Yes

Reviewer #2: Yes

5. Is the manuscript presented in an intelligible fashion and written in standard English?

Reviewer #1: No

Reviewer #2: Yes

6. Review Comments to the Author

You may also provide optional suggestions and comments to authors that they might find helpful in planning their study.

Reviewer #1: In this study protocol, a multicenter, prospective two-arm randomized controlled trial is being proposed to compare two methods of patient education prior to breast cancer therapy. The CINDERELLA APProach is an artificial intelligence (AI) system that will be used as the intervention arm. The primary objectives will be (1) to compare the differences in patient’s agreement of expectations and aesthetic outcome in the two arms and (2) in the intervention arm only, evaluate the agreement of the AI-evaluation and the patient’s post-therapy self-evaluation.

Minor revisions:

1- Page 5: If block randomization will be used, state the block size.

2- Table 1: The primary and secondary objectives are not stated clearly. Use complete sentences when describing each objective.

3- State the software that will be used to store the aggregate data.

4- Indicate if all statistical analyses will be conducted with the R package.

5- Thoroughly proofread the document. Many grammatical errors are present.

6- To assist in the review process, add line numbering to the document.

Reviewer #2: This prospective randomized multcentre trial aims to improve shared decision making for breast cancer patients undergoing locoregional therapy through an artificial-intelligence cloud-based platform (CINDERELLA platform). The goal of this platform is to improve patient satisfaction, psychosocial wellbeing, and HRQoL through this platform. This platform is innovative and could allow for a cost-effective, widely available tool to improve patient knowledge of locoregional therapy options prior to surgery.

1. In the introduction, the authors state that this new platform will serve to function as a shared decision making tool regarding locoregional therapy. However, none of the outlined objectives seem to evaluate patient decision making. For patients with the option of breast conservation surgery vs. mastectomy, it seems that the patient will be provided one "matched picture" of the surgical procedure she "chooses" rather than multiple images of her different options for locoregional therapy. Will patients be given the option to view more than one "matched picture" (i.e. breast conservation surgery and mastectomy)? If so, will you evaluate whether this tool helped with their decision making?

2. What efforts will be made or already exist within the platform to ensure that women of different races/ethnicities will be adequately represented within the platform?

3. The first and second primary objectives are to (1) better match patient expectations before and after treatment and (2) assess aesthetic outcome between AI evaluation and self-evaluation after treatment. Multiple studies have demonstrated that women with lower baseline satisfaction with breasts will continue to have lower satisfaction with their breasts compared to their peers. How do you plan to account for this?

4. Minor grammatical errors to correct for manuscript:

-Page 4, Line 6: are be should be changed to may be

-Page 9, Line 4: "of" should be deleted before the word variability

-Page 12, Line 15: aligned should be changed to alignment

7. PLOS authors have the option to publish the peer review history of their article (what does this mean?). If published, this will include your full peer review and any attached files.

Reviewer #1: No

Reviewer #2: No

---

## [Author Response · Author response to Decision Letter 0]

3 Jun 2023

26.5.2023

Dear Editor,

Thank you for allowing us to revise our manuscript. Please find below the response to reviewer comments. Our reply is in RED font. We hope you will find our manuscript suitable for publication in your journal.

With kind regards,

Dr. Orit Kaidar – Person, on behalf of all authors.

Reviewer #1: In this study protocol, a multicenter, prospective two-arm randomized controlled trial is being proposed to compare two methods of patient education prior to breast cancer therapy. The CINDERELLA APProach is an artificial intelligence (AI) system that will be used as the intervention arm. The primary objectives will be (1) to compare the differences in patient’s agreement of expectations and aesthetic outcome in the two arms and (2) in the intervention arm only, evaluate the agreement of the AI-evaluation and the patient’s post-therapy self-evaluation.

1- Page 5: If block randomization will be used, state the block size 

Prospective randomized international multicentre parallel group trial conducted in 5 clinical centres, each attaining an institutional ethics approval. All patients will need to sign a written informed consent form. Randomization will be performed centrally, 1:1 to control arm (conventional current clinical practice for patient education prior to therapy) or intervention arm. Randomization will follow the minimization method proposed by Pocock and Simon (11, 12), a covariate-adaptive randomization procedure that achieves balance within covariate margins. The randomization procedure requires randomization lists of two types: (1) 50:50 lists, with blocks of size two will be used to assign the patients when balance between groups is attained during the randomization process. (2) Unbalanced 20:80 and 80:20 randomization lists in blocks of size five will be used to assign the patients otherwise (Figure 1).

Answer to reviewers: we thank the reviewer for the comment. This section was added to the methods. 

2- Table 1: The primary and secondary objectives are not stated clearly. Use complete sentences when describing each objective. 

Response to reviewer: We thank the reviewer for the comment, the following was revised:

Table 1: CINDERELLA trial - objectives

TRIAL OBJECTIVES

Primary objectives: 

1. To evaluate the difference between both arms of the trial, regarding the match of expectations about the aesthetic outcome before and after treatment.

2. Experimental arm only, the agreement of the pre-treatment AI-evaluation (output) and patient’s post-therapy self-evaluation.

Secondary objectives: 

1. To evaluate the difference, between both arms of the trial, in patient’s body image satisfaction after surgery.

2. To evaluate the difference between both arms of the trial, in health resource consumption (time spent in hospital, number of appointments, duration until treatment additional care sought by patients).

3. To evaluate the difference, between both arms of the trial, in patient’s satisfaction with their professional life.

4. To evaluate the difference between both arms of the trial, in patient’s satisfaction with their sexual life.

5. To evaluate the difference, between both arms of the trial, in patient’s satisfaction with their general health-related quality of life.

3- State the software that will be used to store the aggregate data. 

Reply to reviewer: We added the following to the text:

Data will be stored in the ISO27001 certified CANKADO Software. All clinical centres received institutional security approval for data safety of the whole trial process.

4- Indicate if all statistical analyses will be conducted with the R package – 

Answer to reviewers

All statistical analysis will be conducted using R statistical software.

5- Thoroughly proofread the document. Many grammatical errors are present 

Response to reviewer: thank you for your comment. The manuscript was reviewed by an English native, the changes are in red font. 

6- To assist in the review process, add line numbering to the document. - Done

Reviewer #2: This prospective randomized multcentre trial aims to improve shared decision making for breast cancer patients undergoing locoregional therapy through an artificial-intelligence cloud-based platform (CINDERELLA platform). The goal of this platform is to improve patient satisfaction, psychosocial wellbeing, and HRQoL through this platform. This platform is innovative and could allow for a cost-effective, widely available tool to improve patient knowledge of locoregional therapy options prior to surgery.

1. In the introduction, the authors state that this new platform will serve to function as a shared decision making tool regarding locoregional therapy. However, none of the outlined objectives seem to evaluate patient decision making. For patients with the option of breast conservation surgery vs. mastectomy, it seems that the patient will be provided one "matched picture" of the surgical procedure she "chooses" rather than multiple images of her different options for locoregional therapy. Will patients be given the option to view more than one "matched picture" (i.e. breast conservation surgery and mastectomy)? If so, will you evaluate whether this tool helped with their decision making? 

We thank the reviewer for the pertinent comment.

The indication for a certain type of surgery or more will be discussed with the patient at the surgical appointment. More than one type of surgery can be discussed with the patient. Patient will be provided with matching pictures for each of the surgery types that were discussed - more than one if it is the case. The change in choice will be recorded although this is not a primary outcome of the trial.

This text was added to the introduction:

The interventional arm will have access to the CINDERELLA platform, that provides information about the locoregional therapy and the possible aesthetic outcomes (visual with a scale) according to the different type of surgeries the patient was offered. The types of surgeries offered, and patient’s choice will be recorded for both groups. Patients’ expectation and satisfaction with the outcomes will be recorded. 

2. What efforts will be made or already exist within the platform to ensure that women of different races/ethnicities will be adequately represented within the platform?

We appreciate the reviewer comment on the topic. This is a very actual and important issue that was emphasized by the EY. Our discussions for designing the protocol and the app ethics committee representatives. The app was design to allow different level of information according to different level of education (per personas). We are aware that the CINDERELLA is a EU funded project and due to that we have a limited choice in terms of partners and consequently races/ethnicities. Nevertheless the inclusion of diverse countries from both western and eastern Europe and also Israel will help us to enrich the platform with a more diversified set of women. We aim also at allow external non funded partners to join us not only to improve patient accrual but also to increase the diversity of the dataset. Both technology partners are committed to improve the app in the future to assure its generality. 

We added this section to the discussion:

The CINDERELLA platform is planned to have different levels of information to match different level of education and needs of the patients. The AI system might be limited to races/ethnicities of patients from the clinical sites that are involved in the project. The project is designed to have input from different stakeholders to upgrade/improve/adjust the platform to patient’s needs. The partners aim to involve non funded partners to join the project to increase the diversity of the dataset and to have representation of all races/ethnicities to assure that the platform serves all.

3. The first and second primary objectives are to (1) better match patient expectations before and after treatment and (2) assess aesthetic outcome between AI evaluation and self-evaluation after treatment. Multiple studies have demonstrated that women with lower baseline satisfaction with breasts will continue to have lower satisfaction with their breasts compared to their peers. How do you plan to account for this?

Response to reviewer: Thank you for your comment, the question is very relevant and that is one of the reasons we choose the expectation match between the before and after the surgery. Using this type of metric we will always have the possibility to overcome this problem. Additionally the use of QOL questionnaires will help us track the lower baseline cases and track them accordingly

4. Minor grammatical errors to correct for manuscript: thank you for noticing, we made changes in the text. The manuscript was now reviewed by a native English speaker. 

-Page 4, Line 6: are be should be changed to may be - Done

-Page 9, Line 4: "of" should be deleted before the word variability - Done

-Page 12, Line 15: aligned should be changed to alignment - Done

---

## [Decision Letter · Decision Letter 1]

18 Jul 2023

Evaluating the ability of an artificial-intelligence cloud-based platform designed to provide information prior to locoregional therapy for breast cancer in improving patient’s satisfaction with therapy: The CINDERELLA trial

PONE-D-23-07134R1

Dear Dr. Kaidar-Person,

We’re pleased to inform you that your manuscript has been judged scientifically suitable for publication and will be formally accepted for publication once it meets all outstanding technical requirements.

Kind regards,

Murat Akand, MD, FEBU

Academic Editor

PLOS ONE

Additional Editor Comments (optional):

Reviewers' comments:

Reviewer's Responses to Questions

**Comments to the Author**

1. Does the manuscript provide a valid rationale for the proposed study, with clearly identified and justified research questions?

Reviewer #1: Yes

Reviewer #2: Yes

2. Is the protocol technically sound and planned in a manner that will lead to a meaningful outcome and allow testing the stated hypotheses?

Reviewer #1: Yes

Reviewer #2: Yes

3. Is the methodology feasible and described in sufficient detail to allow the work to be replicable?

Reviewer #1: Yes

Reviewer #2: Yes

4. Have the authors described where all data underlying the findings will be made available when the study is complete?

Reviewer #1: No

Reviewer #2: Yes

5. Is the manuscript presented in an intelligible fashion and written in standard English?

Reviewer #1: Yes

Reviewer #2: Yes

6. Review Comments to the Author

You may also provide optional suggestions and comments to authors that they might find helpful in planning their study.

Reviewer #1: The authors have adequately addressed all comments.

Reviewer #2: All questions were answered fully by the authors of the study. The manuscript should be accepted for publications.

7. PLOS authors have the option to publish the peer review history of their article (what does this mean?). If published, this will include your full peer review and any attached files.

Reviewer #1: No

Reviewer #2: No

---

## [Editor Report · Acceptance letter]

24 Jul 2023

PONE-D-23-07134R1 

Evaluating the ability of an artificial-intelligence cloud-based platform designed to provide information prior to locoregional therapy for breast cancer in improving patient’s satisfaction with therapy: The CINDERELLA trial 

Dear Dr. Kaidar-Person:

I'm pleased to inform you that your manuscript has been deemed suitable for publication in PLOS ONE. Congratulations! Your manuscript is now with our production department. 

Kind regards, 

on behalf of

Dr. Murat Akand 

Academic Editor

PLOS ONE